# Lymphocytes influence *Leishmania major* pathogenesis in a strain-dependent manner

**Md. Abu Musa**[1,2,3], **Risa Nakamura**[1,2,3], **Asma Hena**[2,3], **Sanjay Varikuti**[4], **Hira L. Nakhasi**[5], **Yasuyuki Goto**[6], **Abhay R. Satoskar**[4], **Shinjiro Hamano**[1,2,3]*

**1** Doctoral Leadership Program, Graduate School of Biomedical Sciences, Nagasaki University, Nagasaki, Japan, **2** Department of Parasitology, Institute of Tropical Medicine (NEKKEN), Nagasaki University, Nagasaki, Japan, **3** The Joint Usage/Research Center on Tropical Disease, Institute of Tropical Medicine (NEKKEN), Nagasaki University, Nagasaki, Japan, **4** Department of Pathology, Ohio State University, Columbus, Ohio, United States of America, **5** Division of Emerging and Transfusion Transmitted Diseases, CBER, FDA, Silver Spring, Maryland, United States of America, **6** Laboratory of Molecular Immunology, Graduate School of Agricultural and Life Sciences, The University of Tokyo, Bunkyo-ku, Tokyo, Japan

* shinjiro@nagasaki-u.ac.jp

**Data Availability Statement:** All relevant data are within the manuscript and its Supporting Information files.

## Abstract

Cutaneous leishmaniasis (CL) is the most common form of leishmaniasis and is caused by several species of *Leishmania* parasite. Clinical presentation of CL varies from a self-healing infection to a chronic form of the disease determined by the virulence of infecting *Leishmania* species and host immune responses to the parasite. Mouse models of CL show contradictory roles of lymphocytes in pathogenesis, while acquired immune responses are responsible for host protection from diseases. To reconcile the inconclusive roles of acquired immune responses in pathogenesis, we infected mice from various genetic backgrounds with two pathogenic strains of *Leishmania major*, Friedlin or 5ASKH, and assessed the outcome of the infections. Our findings showed that the genetic backgrounds of *L. major* determine the impact of lymphocytes for pathogenesis. In the absence of lymphocytes, *L. major* Friedlin induced the lowest inflammatory reaction and pathology at the site of infection, while 5ASKH infection induced a strong inflammatory reaction and severe pathology. Lymphocytes ameliorated 5ASKH mediated pathology, while it exacerbated pathology during Friedlin infection. Excess inflammatory reactions, like the recruitment of macrophages, neutrophils, eosinophils and production of pro-inflammatory cytokines, together with uncontrolled parasite growth in the absence of lymphocytes during 5ASKH infection may induce severe pathology development. Taken together our study provides insight into the impact of differences in the genetic background of *Leishmania* on CL pathogenesis.

## Author summary

Cutaneous leishmaniasis is caused by different species and sub-species of the intracellular parasite *Leishmania*. It is prevalent mainly in tropical and sub-tropical parts of the world. Disease manifestations range from self-healing cutaneous lesions to chronic form of the disease, depending on the infecting species of *Leishmania* and host immune protection. The mechanisms of pathogenesis are largely unknown. Lymphocytes play a central role in

**Funding:** This work was supported by the Joint Usage/Research Center on Tropical Disease, Institute of Tropical Medicine (NEKKEN), Nagasaki University (to RN, SH), the Global Health Innovative Technology (GHIT) Fund, Japan [G2015-115] (to AS, HN, SH), the Global Leadership Program, Nagasaki University (to AM, RN, SH) and JSPS KAKENHI [18H02649](to YG). The funders had no role in study design, data collection and analysis, decision to publish, or preparation of the manuscript.

**Competing interests:** The authors have declared that no competing interests exist.

the protection against *Leishmania* infection; however, their role in pathogenesis is poorly defined. Experimental infection studies showed the inconsistent role of lymphocytes in pathogenesis. Here, we compared disease outcomes in mice infected with different strains of *Leishmania major*, either Friedlin or 5ASKH. The pathogenesis caused by *L. major* 5ASKH infection was suppressed by the lymphocytes, while it was augmented by the lymphocytes during *L. major* Friedlin infection. Thus we found that the influence of lymphocytes in pathogenesis was determined by the genetic background of the parasites.

## Introduction

Leishmaniasis is a group of diseases caused by protozoan parasites belonging to *Leishmania* species. Leishmaniasis threatens more than 350 million people from 98 countries in tropical and subtropical regions of the world with an estimated 0.7–1 million new cases and 20,000 to 30,000 deaths annually [1]. The disease can be categorized into three major forms: cutaneous, mucocutaneous and visceral leishmaniasis [2]. Cutaneous leishmaniasis (CL) is the most common form of the disease. Clinical manifestations of CL vary from asymptomatic to a non-healing chronic form of the disease determined by the infecting *Leishmania* species and by the host immune response to the parasite [3, 4]. The biological processes that may help to explain why some cases are asymptomatic and others show a variety of clinical pathologies, across the spectrum of self-healing to severe forms of leishmaniasis, are ill-defined. The diverse clinical manifestations of CL demand a thorough investigation of the pathophysiology of the infection in varying host defenses and parasite virulence.

Mouse models of leishmaniasis have been extensively studied to understand and characterize the host-parasite interactions during infections [5]. In mouse models of CL, the mechanism of resistance to *Leishmania major* infection has been well established using genetically resistant C57BL/6 mice that reproduce the self-cure disease outcomes typically seen in humans [3]. Resistance to *L. major* infection is mainly mediated by the CD4$^+$ and CD8$^+$ T lymphocytes, represented by the induction of Th1 immune response and IFN-γ dependent killing of the parasite by macrophages [5, 6]. Conversely, *L. major* infection drives BALB/c mice towards a Th2 response that allows for parasite replication and persistence [5].

While the role of the acquired immune response in protection against leishmaniasis has been well documented, its role in pathogenesis is not fully understood. Experimental CL studies showed the discrepant role of T/B lymphocytes in disease pathology. Some studies showed that lesion development was independent of T/B cells in CL models using *L. major* [7] or *Leishmania amazonensis* [8]. Other studies showed that lesion development caused by *L. major* [9, 10] or *L. amazonensis* [11] infection was delayed and less severe in the absence of T/B cells. Soong *et al.* showed that CD4$^+$ T lymphocytes are indispensable for pathogenesis during *L. amazonensis* infection [12]. In these studies, different species or isolates of *Leishmania* parasite were used. The discrepant roles of T/B cells in pathogenesis may be due to the differences in parasite species or strains. It is reasonable to hypothesize that the genetic background of the *Leishmania* strain is crucial to determine the variable influence of lymphocytes for pathology development.

To investigate our hypothesis, we infected mice of various background with two strains of *L. major* with different degrees of pathogenicity, namely strains Friedlin and 5ASKH, and measured the disease outcome. Both *L. major* Friedlin and 5ASKH infections were controlled in resistant C57BL/6 mice with transient lesion development and limiting parasite burden. Susceptible BALB/c mice showed progressive lesion development with uncontrolled parasite

growth. In the absence of T/B lymphocytes, 5ASKH strain caused more progressive lesion development, while Friedlin strain caused limited lesion development whilst maintaining similar levels of parasite burden to 5ASKH strain. Our results indicated that lymphocytes suppressed pathogenesis during 5ASKH infection, while the presence of lymphocytes during Friedlin infection upregulated pathogenesis. We conclude that the genetic background of *L. major* determines the distinct influence of lymphocytes for the pathogenesis of this organism.

## Materials and methods

### Ethics statement

The study protocol was approved by the Committee for Ethics on Animal Experiments (approval number 1505181226, 1505181227) and Recombinant DNA experiments (1403041262) in Nagasaki University. All of the studies were conducted under the guidelines for animal experiments, Nagasaki University and according to Japanese law for Humane Treatment and Management of Animals (Law No. 105 dated 19 October 1973 modified on 2 June 2006).

### Animals

Six to eight weeks old female mice were used in all experiments. In this study, we included wild-type (WT) and nude BALB/c, WT and recombination activating gene 2 (Rag2) knockout (KO) C57BL/6, and NOD-SCID mice. WT BALB/c (BALB/cCrSlc) and C57BL/6 (C57BL/6JJmsSlc) mice were purchased from SLC, Japan. Nude BALB/c (CAnN.Cg-Foxn1$^{nu}$/CrlCrlj) and NOD-SCID (NOD.CB17-Prkdc scid/J) mice were purchased from Charles River Laboratories, Japan. Rag2 KO C57BL/6 (B6(Cg)-Rag2tm1.1Cgn/J) mice were purchased from the Jackson Laboratory. Experimental mice were maintained under specific pathogen-free conditions at the Animal Facilities of Nagasaki University. Experimental animals were fed with standard rodent pellets supplemented with grain and given water ad libitum. Throughout this study, the animals were housed in a climate-controlled (23±2˚C; relative humidity, 60%) and photoperiod controlled (12-hours light-dark cycles) animal quarters. Only healthy mice were included in the study.

### *Leishmania* parasite and infection

*L. major* clone V9 (MHOM/IL/80/Friedlin) and *L. major* (MHOM/SU/73/5-ASKH) strain abbreviated as *L. major* Friedlin and *L. major* 5ASKH respectively were used in this study. The parasite culture was maintained in medium 199 (Gibco) supplemented with 10% heat-inactivated fetal bovine serum (CCB, Japan), 100 U/ml penicillin and 100 μg/ml streptomycin (Gibco) at 26˚C in a biological incubator. Cultures were passaged to fresh medium at a 50-fold dilution in every 2–3 days. Stationary phase promastigotes were harvested and suspended in 1× PBS. Mice were infected with $5×10^6$ stationary phase promastigotes of *L. major* Friedlin or 5ASKH by subcutaneous injections into their right hind footpad. Following infection, footpad thickness was measured weekly using an analog caliper. Footpad swelling was calculated by subtracting the thickness of the uninfected counter footpad from the thickness of the infected footpad. Parasite burdens in the footpads and popliteal lymph nodes (pLN) were quantified at 6 weeks post-infection. Parasite burdens in the footpads and draining lymph nodes were quantified by either limiting dilution assay (LDA) or homogenizing tissue in 10 ml of complete medium 199 and incubated in 25cm$^2$ culture flasks at 26˚C for the detection of parasites transformed in vitro from amastigotes to motile promastigotes. At 3 days of culture, promastigotes were counted microscopically. For LDA tissues were homogenized in 1 ml complete medium

199 following 2 fold serial dilution across 96-well culture plates. Culture plates were incubated at 26˚C. At 1-week culture plates were observed under a microscope for the presence of motile promastigotes.

## Isolation of cells from the footpad of mouse

Mice were euthanized and the footpads have collected aseptically. The footpads were cut tangentially to the bone and mechanically disrupted with surgical scissors in 5 ml cold RPMI medium supplemented with 1 mg/ml collagenase (WAKO, Japan) and 1 mg/ml DNase (Roche Diagnostics, Japan). Disrupted footpads were then incubated for 30 min in a 37˚C water bath with continuous shaking. Following incubation, the reactions were stopped by adding cold RPMI medium with 2.5 mM EDTA and strained through a 70 μm cell strainer. The cell suspensions were centrifuged at 440g for 6 min at 4˚C and re-suspended in complete RPMI medium supplemented with 10% heat-inactivated fetal bovine serum, 100 U/ml penicillin and 100 μg/ml streptomycin.

## Flow cytometric analysis

Footpad cells were stained separately with 2 sets of fluorescence-conjugated antibody cocktail. The first set of antibody cocktail consisted of PE-conjugated anti-CD86, APC-conjugated anti-MHC class-II, FITC-conjugated anti-CD11b, PE-Cy7-conjugated anti-F4/80, APC-Cy7-conjugated anti-CD45, and PerCp-Cy5.5-conjugated anti-CD11c. The second set of the antibody included PE-conjugated anti-SiglecF, APC-conjugated anti-Gr1, FITC-conjugated anti-CD11b, PE-Cy7-conjugated anti-F4/80, APC-Cy7-conjugated anti-CD45, and PerCp-Cy5.5-conjugated anti-NKp46 antibody. Cells were then analyzed on a FACSVerse™ (BD Bioscience, NJ, USA). Data were analyzed with FlowJo software v10.

## Ex-vivo cytokine analysis

Cells isolated from mouse footpad were plated at the concentration of $5 \times 10^5$ cells per well in 96-well culture plates in 200 μl RPMI medium (supplemented with 10% heat-inactivated fetal bovine serum, 100 U/ml penicillin and 100 μg/ml streptomycin). After 72h of PMA (40 ng/ml)/Ionomycin (4 μg/ml) stimulation, culture supernatants were stored at -30˚C. Concentrations of IFN-γ, TNF-α, IL-12 p40, and IL-10 were measured by sandwich ELISA as per manufacturer (R&D Systems) instructions.

## Quantitative RT-PCR

Mice footpads were homogenized with 1 ml TRIzol (Thermo Fisher Scientific) and φ1.0 stainless steel beads in the 2 ml tube using Micro Smash MS100R (TOMY, Tokyo, Japan) at 4˚C using tissue homogenizer (Tomy, Micro Smash MS-100). RNA was isolated using Trizol Reagent according to manufacturer instructions. RNA yield was measured by a spectrophotometer (Beckman Coulter DU-730). We found 30–53 μg RNA from each sample, and 2 μg of total RNA was used as the template for the synthesis of 20 μl cDNA. Synthesized cDNA was analyzed for mouse TNF-α, IL-1β, IL-6, IL-12 p40, inducible nitric oxide synthase (iNOS), and arginase 1 by reverse transcriptase real-time PCR. PCR method and primer sequences were described before [13]. Briefly real-time polymerase chain reaction (PCR) assay was carried out using 1 μl of cDNA as the template, 10 μl of SYBR Select Master Mix (Thermo Fisher Scientific) and primers on the ABI Prism 7000 Sequence Detection System (Thermo Fisher Scientific). Data were analyzed by 2−ΔΔ Ct methods and normalized by GAPDH. The thermal

cycling conditions for the PCR were 94˚C for 10 min, followed by 40 cycles of 94˚C for 15 s and 60˚C for 1 min.

## Statistical analysis

All statistical analysis was performed using GraphPad Prism Software Version-5. To check data distribution we had performed the Shapiro-Wilk normality test. Statistical analysis of the lesion kinetics was performed with two-way ANOVA to test differences between groups. In all cases, a P-value of less than 0.05 was considered statistically significant.

## Results

### Parasite strain-dependent influence of T lymphocytes on pathogenesis in susceptible BALB/c mice

To characterize the role of T lymphocytes and *L. major* strains on pathogenesis, we infected susceptible BALB/c WT and nude mice with $5 \times 10^6$ stationary phase promastigotes of either *L. major* strain, Friedlin or 5ASKH. Following infections, the lesions were measured weekly, until 6 weeks post-infection. The parasite burden in footpads and pLN were also measured at 6 weeks post-infection, at the end of the experiment. Both *L. major* Friedlin and 5ASKH infection showed progressive lesion development in WT mice (**Fig 1A**), where three of eight mice infected with 5ASKH developed ulcerative lesions compared with one of nine for Friedlin-infected mice (**Table 1**). In nude mice, *L. major* 5ASKH infection caused more severe ulcerative lesions in all 10 mice compared to 3/8 in WT mice (**Fig 1B** and **Table 1**). Conversely, *L. major* Friedlin infection caused minimal lesion development with no ulcer formation in all 11 nude mice and 1/9 in WT mice (**Fig 1B** and **Table 1**). It is worth noting that both Friedlin and 5ASKH showed a similarly high level of parasite burden in the footpads and pLN not only in nude mice but also in WT mice (**Figs 1C and S1**). These results showed that T lymphocytes reduced immunopathology during 5ASKH infection, while T lymphocytes exacerbated lesion development during Friedlin infection.

### Parasite strain-dependent influence of T/B lymphocytes on pathogenesis in resistant C57BL/6 mice

Next, we examined the *L. major* strain-dependent influence of lymphocytes for lesion development in resistant C57BL/6 mice that allows transient *L. major* infection. WT and Rag2 KO C57BL/6 mice were infected with either *L. major* Friedlin or 5ASKH. The WT mice showed transient non-ulcerative lesion development during infection with both *L. major* strains (**Fig 2A and 2B**) and controlled parasite growth (**Fig 2C**). In Rag2 KO mice, which lack both conventional T/B lymphocytes, 5ASKH infection caused delayed but progressive and non-ulcerative lesion development, whereas Friedlin infection caused delayed and lesser lesion development (**Fig 2A and 2B**) despite a similar level of parasite burden in Friedlin- or 5ASKH-infected footpads and pLN (**Figs 2C and S2**).

### *L. major* Friedlin and 5ASKH caused minimum or severe pathology respectively in NOD-SCID mice

Lack of pathogenesis by Friedlin strain in severe immunodeficient condition was also confirmed in NOD-SCID mice. NOD-SCID mice lack T/B cells and also defects in functions of NK, NKT, macrophage, and dendritic cells. In NOD-SCID mice, there was no difference in parasite burden between Friedlin and 5ASKH infected footpad and pLN (**S3C Fig**). However,

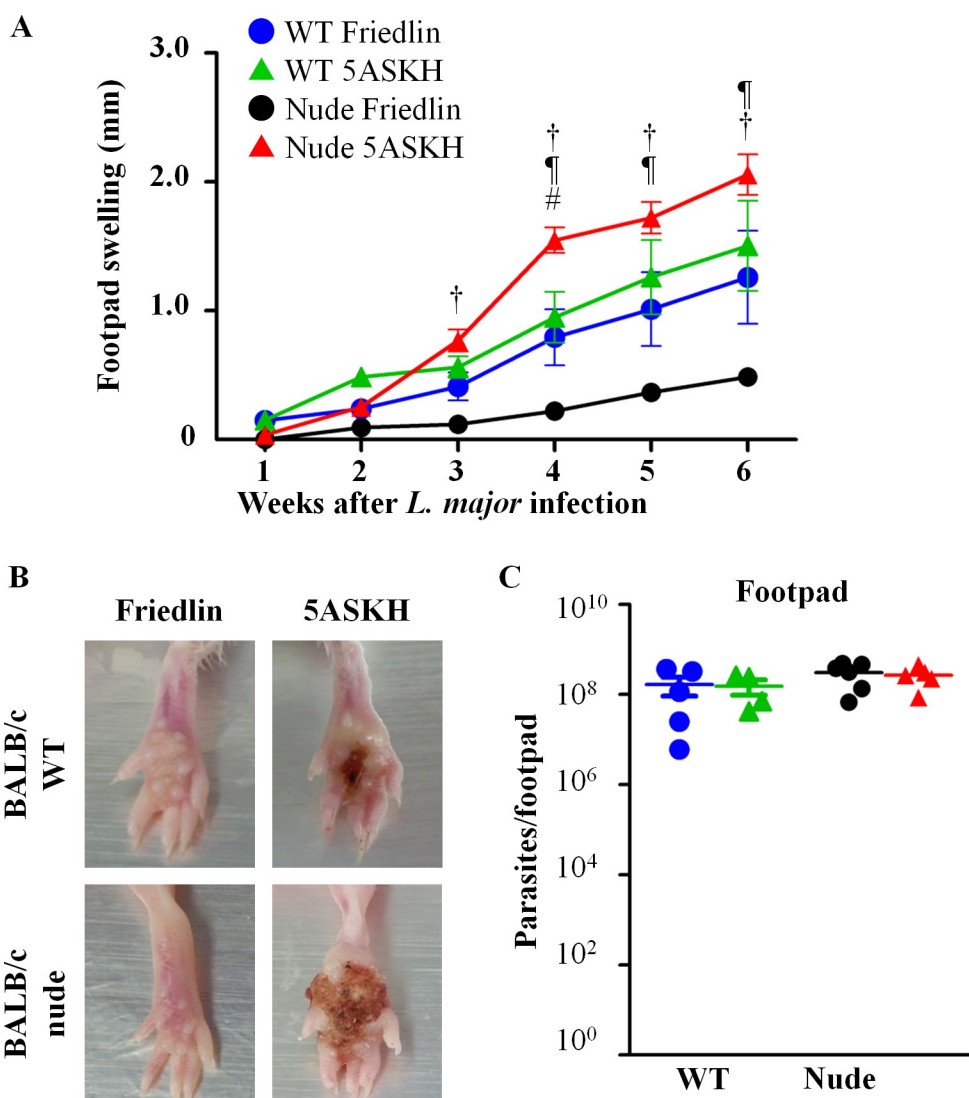

**Fig 1. Parasite strain-dependent influence of T lymphocytes on pathogenesis in susceptible BALB/c mice.** BALB/c WT and nude mice were infected with $5\times10^6$ stationary phase promastigotes of *L. major* Friedlin or 5ASKH subcutaneously into the hind footpad. (**A**) Footpad swelling after *L. major* infection. Each group contains 8–11 mice. All the data are from normal distribution except data of BALB/c mice at 1 and 3 weeks and BALB/c-nude mice at 2 weeks post-infection of *L. major* Friedlin have deviated from normal distribution. Data deviated from normal distribution were excluded from statistical comparison (**B**) Representative photographs of *L. major*-infected footpads at 6 weeks post-infection. (**C**) Parasite burden in footpads at 6 weeks post-infection. Each mouse footpad was homogenized in 10 ml culture media and cultured for 3 days then viable promastigotes were counted microscopically. Each group contains four–six mice. Symbols: BALB/c mice infected with *L. major* Friedlin blue circle or 5ASKH green triangle, BALB/c-nude mice infected with *L. major* Friedlin black circle or 5ASKH red triangle; Data are the mean ± SEM. Statistical analyses for lesion size by ANOVA. *†¶# means P<0.05 between BALB/c-Friedlin×BALB/c-5ASKH: *; Nude-Friedlin×Nude-5ASKH: †; BALB/c-Friedlin×Nude-Friedlin: ¶ and BALB/c-5ASKH×Nude-5ASKH: #.

*L. major* 5ASKH showed non-ulcerative and progressive lesion development, but Friedlin showed almost no lesion development (Table 1 and **S3A and S3B Fig**).

**Table 1. Ulcer formation at 6 weeks post-infection with *L. major* Friedlin or 5ASKH.**

| Mice | Ulcer formation % (number) | |
|---|---|---|
| | **Friedlin** | **5ASKH** |
| BALB/c | 11.1 (1/9) | 37.5 (3/8) |
| BALB/c-nude | 0 (0/11) | 100 (10/10) |
| C57BL/6 | 0 (0/8) | 0 (0/8) |
| C57BL/6-Rag2 KO | 0 (0/5) | 0 (0/3) |
| NOD-SCID | 0 (0/8) | 0 (0/8) |

## Defective infiltration of immune cells to the site of *L. major* Friedlin infection

Our study showed that both parasite and host factors affected the clinical presentation of CL. Inflammatory responses are mediated by immune cells that control not only parasite growth but also contribute to the pathogenesis of CL [3, 14]. To characterize the inflammation induced by either *L. major* Friedlin or 5ASKH infections, we isolated immune cells from Rag2 KO-infected footpads at 4 weeks post-infection and analyzed these cells by flow cytometry. It is worth noting that, *L. major* Friedlin infection failed to recruit immune cells to the site of infection, while a 25 times higher number of cells were recruited to the footpads infected with *L. major* 5ASKH. Among the cells recruited to the site of *L. major* 5ASKH infection, macrophages (CD11b$^+$F4/80$^+$), neutrophils (CD11b$^+$Gr1$^+$), and eosinophils (CD11b$^+$SiglecF$^+$) were the major cell populations. In addition, *L. major* 5ASKH infected footpad also contained a higher proportion and number of activated macrophages (CD11b$^+$F4/80$^+$CD86$^+$) compared to that with *L. major* Friedlin (**Figs 3A and 3B and 4A and 4B and S4**).

## Lower activation of host immune responses by *L. major* Friedlin infection

Immune cells were isolated from the footpads of Rag2 KO mice 4 weeks after infection with *L. major* Friedlin or 5ASKH and were cultured for 72 h with or without PMA-ionomycin stimulation. Culture supernatants were analyzed for IFN-γ, TNF-α, IL-12, and IL-10 cytokines by ELISA. *L major* infection augmented the production of cytokines and *L. major* 5ASKH induced significantly higher cytokine production than Friedlin (**Fig 5**). After stimulation with PMA-ionomycin, a similar trend was observed (**S5 Fig**).

## Discussion

The distinct impact of lymphocytes for the pathogenesis of CL was confirmed to be attributed to the genetic background of *L. major*. Our results clarified the contradictory findings of previous independent studies, in which lesion development by CL was either similar [7, 8], delayed or less pronounced [9, 10, 12] in the absence of T/B lymphocyte-mediated immune responses compared to wild-type mice. The *L. major* 5ASKH infection model was consistent with reports that lesion development by either *L. major* (MHOM/UZ/91/PM2) or *L. amazonensis* (MPRO/BR/72/M1845) was similar or less in the presence of lymphocytes [7, 8]. The *L. major* Friedlin infection model also explains the previous report that delayed or no lesion development in the absence of T/B lymphocytes by *L. major* (WHOM/IR/-/173) [9, 10] or *L. amazonensis* (MHOM/BR/77/LTB0016) [12] infection. Thus, the impact of lymphocytes for pathogenesis can be determined by the genetic background of *L. major*. In this study, the genetic background of *L. major* strains was not analyzed, which is one of the limitations of the study. We compared only two strains of *L. major* we had no comparison with other strains of *L. major* or

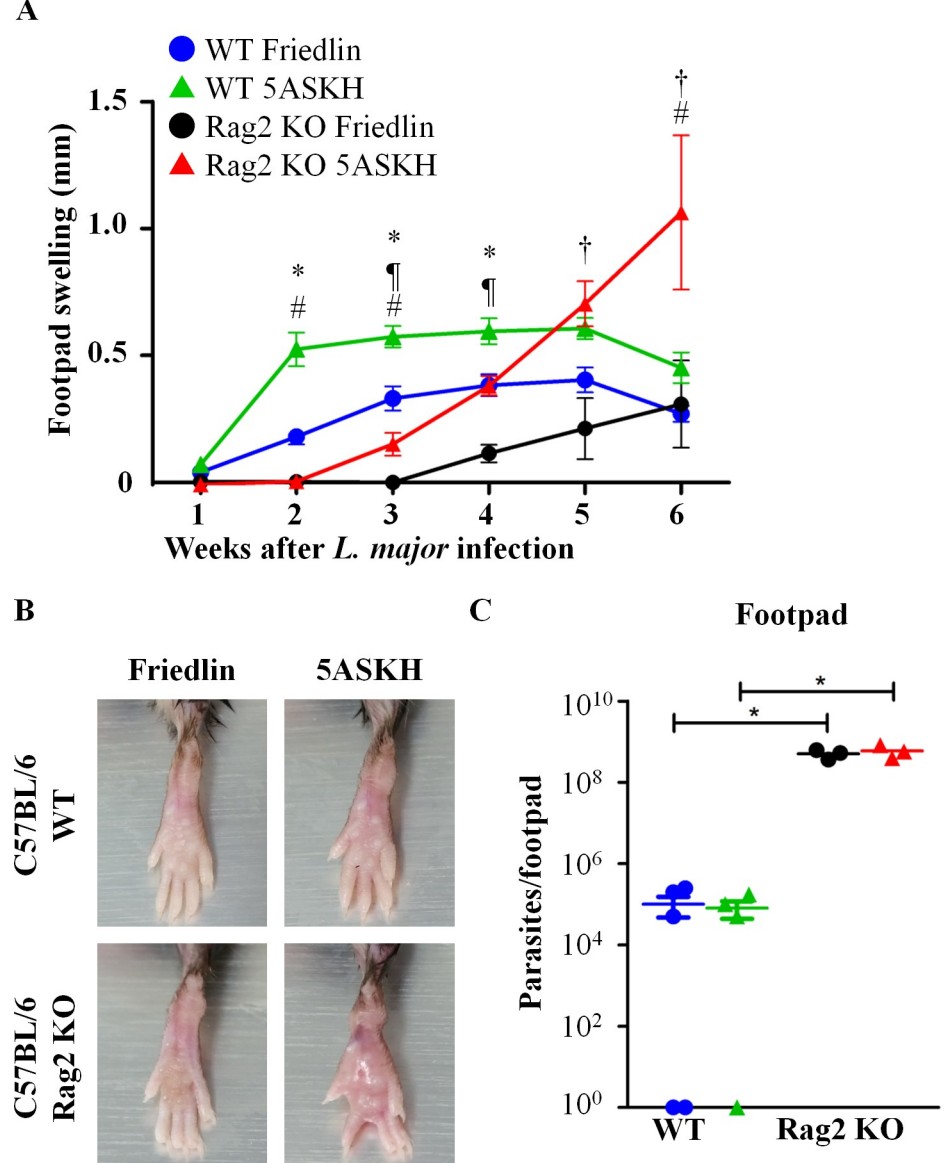

**Fig 2. Parasite strain-dependent influence of T/B lymphocytes on pathogenesis in resistant C57BL/6 mice.** C57BL/6 WT and Rag2 KO mice were infected with $5\times10^6$ stationary phase promastigotes of *L. major* Friedlin or 5ASKH subcutaneously into the hind footpad. (A) Footpad swelling after *L. major* infection. All the data are from normal distribution except data of C57BL/6 mice at 1 week post-infection of *L. major* Friedlin have deviated from normal distribution. Data deviated from normal distribution were excluded from statistical comparison. Symbols: WT mice infected with *L. major* Friedlin blue circle or 5ASKH green triangle, Rag2 KO mice infected with *L. major* Friedlin black circle or 5ASKH red triangle; 3–10 mice per group. (B) Representative photographs of a *L. major*-infected footpad at 6 weeks post-infection. (C) Parasite burden at 6 weeks post-infection. Each mouse footpad was homogenized in 10 ml culture media and cultured for 3 days then viable promastigotes were counted microscopically. Three–five mice per group. Data are the mean ± SEM. Results are representative of two independent experiments with similar outcomes. *†¶# means P<0.05 between WT-Friedlin×WT-5ASKH: *; Rag2KO-Friedlin×Rag2KO-5ASKH: †; WT-Friedlin×Rag2KO-Friedlin: ¶ and WT-5ASKH×Rag2KO-5ASKH: #.

different species of *Leishmania*. Further immunological analyses are needed to inquire into the differential lymphocyte response towards different *Leishmania* strains and species.

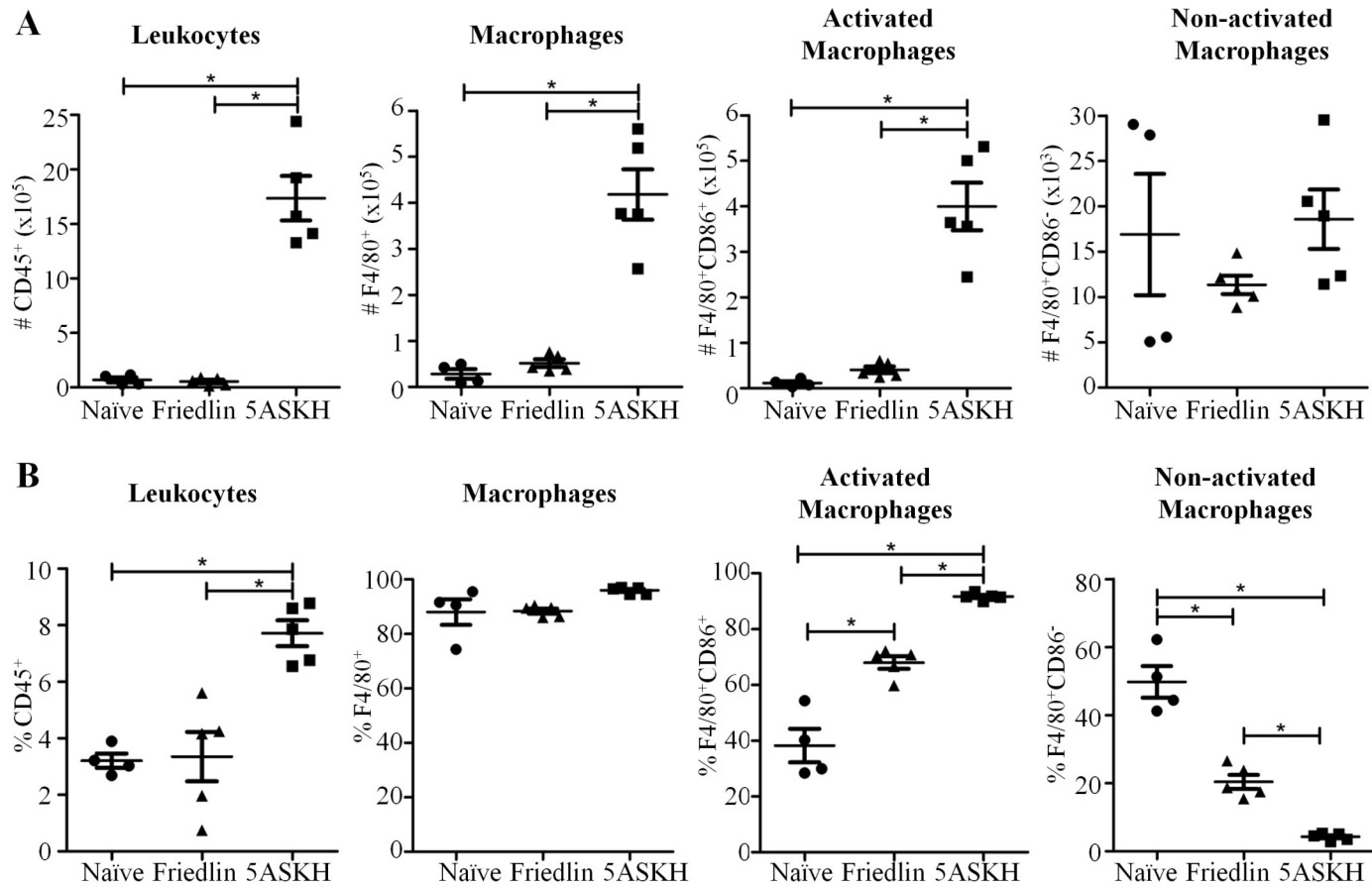

**Fig 3. *L. major* Friedlin infection failed to augment the recruitment of macrophages in Rag2 KO mice.** Rag2 KO C57BL/6 mice were infected with 5×10⁶ stationary phase promastigotes of *L. major* Friedlin or 5ASKH subcutaneously into the hind footpad. At 4 weeks post-infection, cells from the mouse footpads were isolated and analyzed by flow cytometry. The total number (A) and percentages (B) of the cell populations are shown. Four–five mice per group. Data are the mean ± SEM.

In Rag2–deficient C57BL/6 mice, 5ASKH infection caused progressive lesion development. In the early phase of its infection, the degree of footpad swelling was delayed in the absence of T/B lymphocytes indicating that even during 5ASKH infection, lymphocytes could be attributed to lesion formation to some extent. In fact, 5ASKH-infected footpads of WT mice showed a trend of higher expression of pro-inflammatory molecules, such as IL-6, TNF-α, IL-1β, IL-12p40 and iNOS compared with Friedlin-infected footpads (**S6 Fig**). The preceding studies also independently showed delayed lesion development after *L. major* (WHO strain WHOM/IR/-/173) infection in C.B-17 SCID mice compared with BALB/c mice [9, 10].

Pathogenesis can be caused by either the direct tissue-damaging effect of the parasites, parasite-induced immune-mediated skin inflammation or their combined effects. Belkaid *et al.* showed that lesion development starts after the parasite burden reaches its peak and that it coincides with leukocyte trafficking in resistant C57BL/6 mice [15], which emphasized the importance of parasite-induced inflammatory reactions for pathogenesis. We showed that *L. major* 5ASKH infection induced higher inflammatory reactions both in the presence and absence of T/B lymphocytes. Massive recruitment of immune cells, mainly macrophages, neutrophils and eosinophils, was observed at the site of 5ASKH infection even in the absence of lymphocytes. It is consistent with the prior study which showed that lesion development was associated with acute infiltration of leukocytes, primarily macrophages, neutrophils and

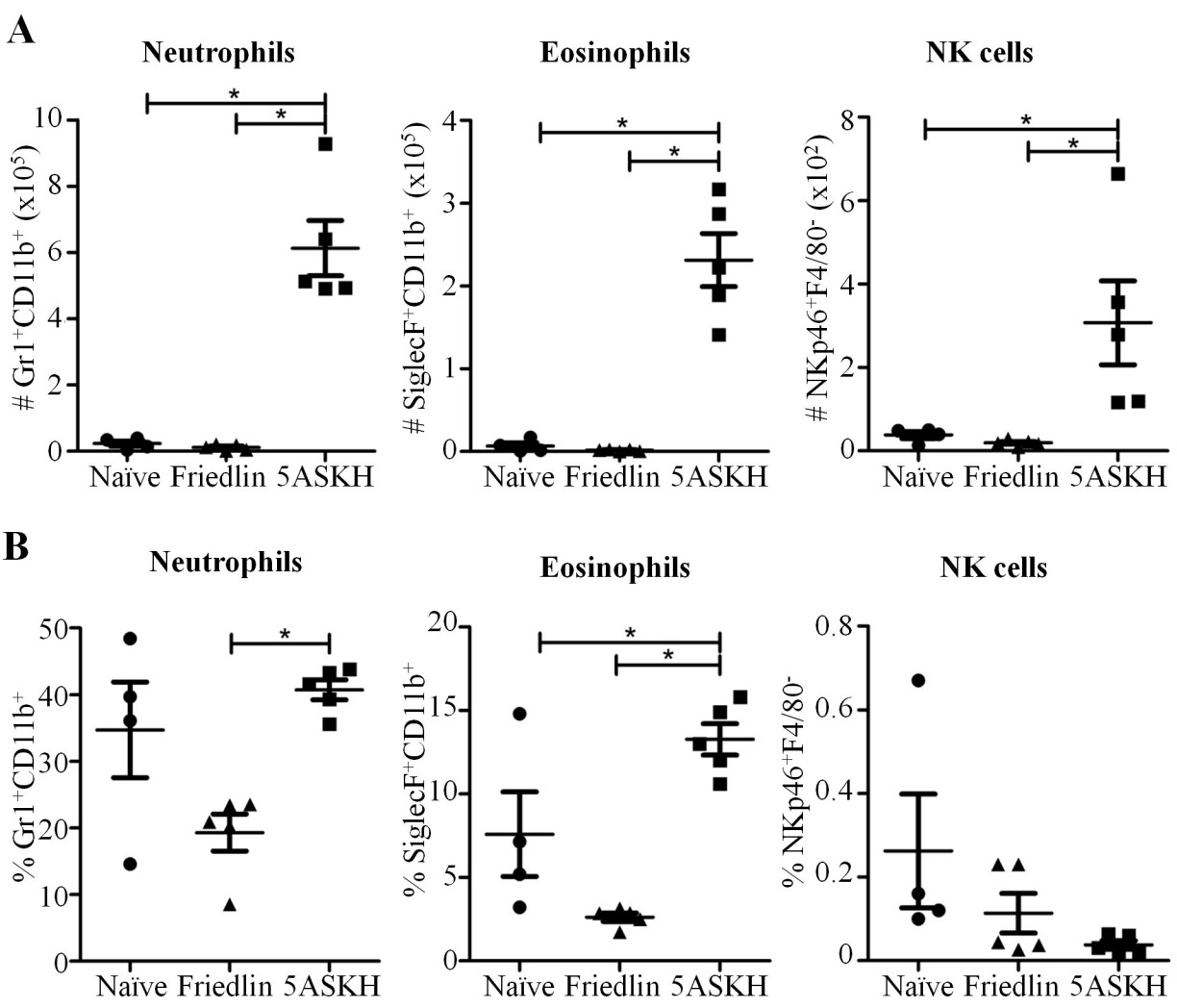

**Fig 4. Defective infiltration of neutrophils, eosinophils, and NK cells during *L. major* Friedlin infection in Rag2 KO mice.** Rag2 KO C57BL/6 mice were infected with $5 \times 10^6$ stationary phase promastigotes of *L. major* Friedlin or 5ASKH subcutaneously into the hind footpad. At 4 weeks post-infection, cells from the mouse footpads were isolated and analyzed by flow cytometry. The total number (A) and percentages (B) of the cell populations are shown. Four–five mice per group. Data are the mean ± SEM.

eosinophils [15]. Immune cells recruited to the site of 5ASKH infection were also highly augmented with a larger number of CD86+-activated macrophages and a higher level of cytokine production was observed compared with Friedlin infection. The inflammatory cytokines are able to activate the macrophages for intracellular killing of the parasite. In wild-type mice, T-lymphocytes are known to be the major source of IFN-γ which is central for the protective immunity. In the absence of T/B lymphocytes, mice failed to control parasites. It indicates that T/B lymphocytes are critical for the control of the parasites, mainly through the production of sufficient amount of critical cytokines, like IFN-γ and TNF-α. In the absence of T/B lymphocytes, macrophages, dendritic cells, NK cells, and group 1 innate lymphoid cells may be one of the sources of IFN-γ, which is insufficient for the protection (**Fig 5**). It may explain why the immunodeficient mice infected with both *L. major* strains showed no difference in parasite burden. However, augmented inflammatory responses after 5ASKH infection may have caused the severe pathology observed. Strong inflammatory reactions at the site of *L. major* 5ASKH infection may explain the severe pathology observed even in the absence of T/B

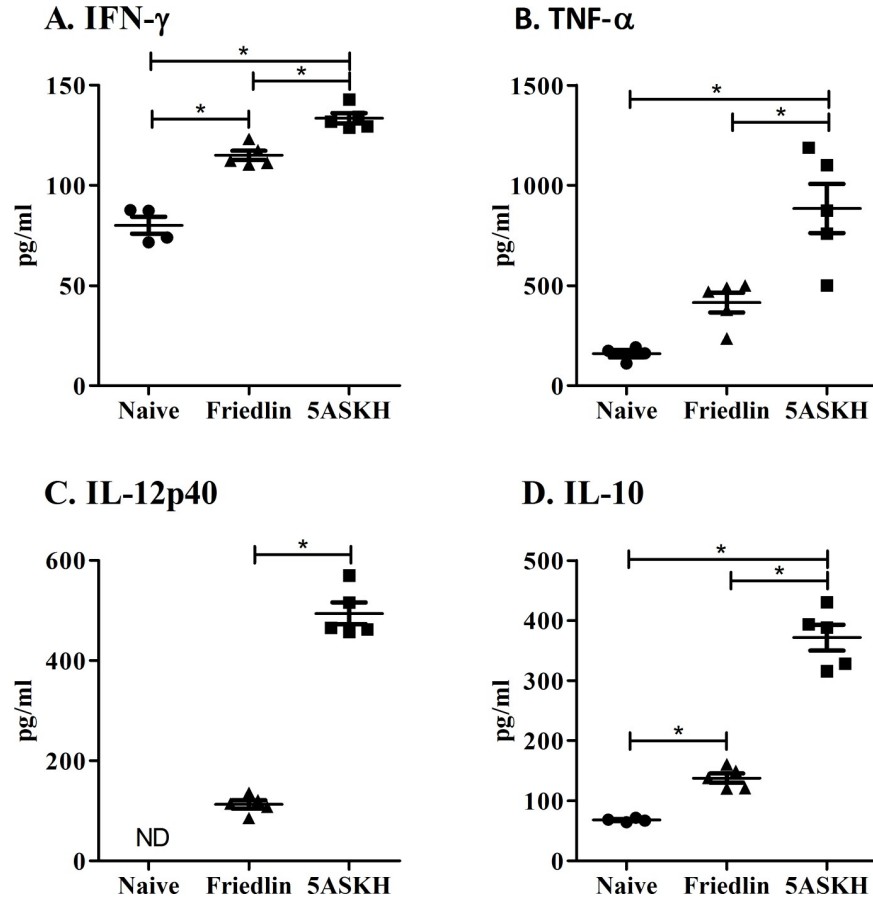

**Fig 5. Lower activation of host immune responses by *L. major* Friedlin infection.** Rag2 KO C57BL/6 mice were infected with 5×10⁶ stationary phase promastigotes of *L. major* Friedlin or 5ASKH subcutaneously into the hind footpad. At 4 weeks post-infection, cells from mouse footpads were isolated and cultured for 3 days. Culture supernatants were analyzed for (A) IFN-γ, (B) TNF-α, (C) IL-12p40, and (D) IL-10 cytokines by ELISA. Three–four mice per group. Data are the mean ± SEM. *, P<0.05.

lymphocytes. *L. major* Friedlin infection failed to induce inflammatory reactions, like recruitment of immune cells and pro-inflammatory cytokines production at the site of infection in the absence of T/B lymphocytes, which may explain the minimum pathology observed. The outcome of Friedlin infection in the immunodeficient mice was consistent with the reported no pathology associated with less infiltration of monocytes at the site of *L. amazonensis* infection in mice lacking T cells [12]. Although we compared inflammatory responses only at 4 weeks after *L. major* Friedlin/5ASKH infection, it is also important to compare early inflammatory responses to address the mechanisms responsible for the different disease outcomes. Different *Leishmania* parasites may have diverse molecules or strategies determining the impact of lymphocytes for immuno-pathogenesis. Comparative analysis showed a significant number of virulence factors or molecules differentially expressed by *Leishmania* isolates [16–18]. We also analyzed only a few cytokines or host molecules after *L. major* infection. Further studies are required to identify the molecules and pathways involved.

CL has a wide spectrum of clinical manifestations depending on the host immunity and the infecting *Leishmania* species. CL is caused by several species of *Leishmania* parasite. In the new world, *L. mexicana*, *L. amazonensis*, *L. panamensis*, *L. brazilienesis*, *L. guyanensis*, *L peruviana* and *L. venezuelensis* are the etiological agents of CL, whereas in the old world *L. tropica*,

*L. killicki*, *L. aethiopica*, *L. arabica*, *L. infantum* and *L. major* are responsible for CL [19, 20]. Many studies have been reported on CL, but few studies focused on the diversity of the clinical presentations. We extensively studied infection outcomes caused by two different strains of *L. major* in animal models with different genetic backgrounds and varying immune competency. *L. major* Friedlin and 5ASKH were originally isolated from patients with CL in the old world, Israel and Turkmenistan respectively, then adapted to the conditions in the Laboratory and established as the strains [21]. In immune-competent mice, we observed different infection outcomes in different host genetic backgrounds. Friedlin infection caused almost no ulceration in any genetic background studied, whereas 5ASKH infection caused ulceration in BALB/c mice and non-ulcerative lesions in C57BL/6 mice. 5ASKH infection caused more ulcerative lesions in T lymphocyte-deficient BALB/c mice compared with WT mice. Terabe *et al.* showed that T/B lymphocytes are a prerequisite for ulcer formation during *L. amazonensis* infection in BALB/c mice [8]. Therefore, particular *Leishmania* species may have distinct immunopathologies of ulcer formation.

## Conclusion

Different species or strains of *Leishmania* were associated with a wide spectrum of diseases in human [19, 20, 22, 23] and mouse models. Previous studies showing that *L. major* LV39 caused a non-healing infection [24–26] whereas other *L. major* strain (MHOM/IL/81/FEBNI) caused healing infections in IL-4 deficient BALB/c mice [25–26]. Furthermore, it now appears that strains of *L. major* can cause variant pathogenesis even in the absence of lymphocytes. Lymphocytes differentially regulate pathogenesis during *L. major* Friedlin/5ASKH infection. *Leishmania* has mechanisms to evade host immune protection, thereby aiding survival inside the host [27]. *L. major* Friedlin may have a superior immune evasion mechanism compared with strain 5ASKH that allows it to survive within the host by eliciting minimum immune-mediated pathogenesis. Understanding the basis for the diversity in pathogenesis is important in determining how to apply our knowledge for the development of new approaches for the treatment of leishmaniasis.

## Supporting information

**S1 Fig. Parasite strain-dependent influence of T lymphocytes on pathogenesis in susceptible BALB/c mice.** BALB/c WT and nude mice were infected with $5\times10^6$ stationary phase promastigotes of *L. major* Friedlin or 5ASKH subcutaneously into the hind footpad. Parasite burden in popliteal lymph nodes (pLN) at 6 weeks post-infection. Each mouse footpad was homogenized in 10 ml culture media and cultured for 3 days then viable promastigotes were counted microscopically. Each group contains 8–11 mice. Symbols: BALB/c mice infected with *L. major* Friedlin ○ or 5ASKH △, BALB/c-nude mice infected with *L. major* Friedlin ● or 5ASKH ▲ Data are mean ± SEM.
(TIF)

**S2 Fig. Parasite strain-dependent influence of T/B lymphocytes on pathogenesis in resistant C57BL/6 mice.** C57BL/6 WT and Rag2 KO mice were infected with $5\times10^6$ stationary phase promastigotes of *L. major* Friedlin or 5ASKH subcutaneously into the hind footpad. Parasite burden in popliteal lymph nodes (pLN) at 6 weeks post-infection. Each mouse footpad was homogenized in 10 ml culture media and cultured for 3 days then viable promastigotes were counted microscopically. 3–9 mice per group. Symbols: wild-type mice infected with *L. major* Friedlin ○ or 5ASKH △, Rag2 KO mice infected with *L. major* Friedlin ● or 5ASKH ▲. Data are mean ± SEM. Results are representative of 2 independent experiments with a similar

outcome.
(TIF)

**S3 Fig. *L. major* Friedlin or 5ASKH infection in NOD-SCID mice.** NOD-SCID mice were infected with $5\times10^6$ stationary phase promastigotes of *L. major* Friedlin ● or 5ASKH ▲ subcutaneously into the hind footpad. (**A**) Footpad swelling after *L. major* infection. Symbols: NOD-SCID mice infected with *L. major* Friedlin  or 5ASKH ; 8 mice per group. (B) Representative photographs of *L. major* infected footpad at 6 weeks post-infection. (C) Footpad parasite burden at 6 weeks post-infection. Each mouse footpad was homogenized in 10 ml culture media and cultured for 3 days then viable promastigotes were counted microscopically. 4–8 mice per group. Data are mean ± SEM. Results are representative of two independent experiments with a similar outcome.
(TIF)

**S4 Fig. Flow cytometric analysis of mouse footpad cells.** Rag2 KO C57BL/6 mice were infected with $5\times10^6$ stationary phase promastigotes of *L. major* Friedlin or 5ASKH subcutaneously into the hind footpad. At 4 weeks post-infection, cells were isolated from mouse footpads and analyzed by flow cytometry. Gating strategies for (A) macrophages; (B) neutrophils, eosinophils, and NK cells.
(TIF)

**S5 Fig. Less activation of host immune responses by *L. major* Friedlin infection.** Rag2 KO C57BL/6 mice were infected with $5\times10^6$ stationary phase promastigotes of *L. major* Friedlin or 5ASKH subcutaneously into the hind footpad. At 4 weeks post-infection cells were isolated from mouse footpads and stimulated with PMA-ionomycin for 3 days. Culture supernatants were analyzed for cytokines by ELISA. 3–4 mice per group. Data are mean ± SEM. *, P<0.05.
(TIF)

**S6 Fig. *L. major* Friedlin infection induced a lower level of the immune response.** C57BL/6 mice were infected with $5\times10^6$ stationary phase promastigotes of *L. major* Friedlin or 5ASKH subcutaneously into the hind footpad. At 4 weeks post-infection total RNA was isolated from mouse footpads and analyzed for mRNA of targeted molecules by real-time RT-PCR. (A) Footpad parasite burden at 4 weeks post-infection measured by limiting dilution assay. (B) mRNA expression in the infected footpad at 4 weeks post-infection. 3–4 mice per group. Data are mean ± SEM.
(TIF)

## Acknowledgments

We would like to offer special thanks to all members of the Department of Parasitology, Institute of Tropical Medicine (NEKKEN), Nagasaki University. We also thank Kate Fox, DPhil, from Edanz Group (www.edanzediting.com/ac) for editing a draft of this manuscript.

## Author Contributions

**Conceptualization:** Md. Abu Musa, Shinjiro Hamano.

**Data curation:** Md. Abu Musa, Asma Hena.

**Formal analysis:** Md. Abu Musa, Risa Nakamura.

**Funding acquisition:** Yasuyuki Goto, Abhay R. Satoskar, Shinjiro Hamano.

**Investigation:** Md. Abu Musa, Risa Nakamura, Yasuyuki Goto, Shinjiro Hamano.

**Methodology:** Md. Abu Musa, Risa Nakamura, Sanjay Varikuti, Yasuyuki Goto, Shinjiro Hamano.

**Resources:** Yasuyuki Goto, Shinjiro Hamano.

**Supervision:** Risa Nakamura, Hira L. Nakhasi, Yasuyuki Goto, Abhay R. Satoskar, Shinjiro Hamano.

**Validation:** Risa Nakamura, Shinjiro Hamano.

**Visualization:** Md. Abu Musa.

**Writing – original draft:** Md. Abu Musa.

**Writing – review & editing:** Md. Abu Musa, Hira L. Nakhasi, Yasuyuki Goto, Shinjiro Hamano.

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
