## [Decision Letter · Decision Letter 0]

5 Aug 2019

Dear Prof. Hamano:

Thank you very much for submitting your manuscript "Leishmania major strain dependent requirement of lymphocytes for the pathogenesis" (#PNTD-D-19-01017) for review by PLOS Neglected Tropical Diseases. Your manuscript was fully evaluated at the editorial level and by independent peer reviewers. The reviewers appreciated the attention to an important problem, but raised some substantial concerns about the manuscript as it currently stands. These issues must be addressed before we would be willing to consider a revised version of your study. We cannot, of course, promise publication at that time.

We therefore ask you to modify the manuscript according to the review recommendations before we can consider your manuscript for acceptance. Your revisions should address the specific points made by each reviewer. 

When you are ready to resubmit, please be prepared to upload the following:

(1) A letter containing a detailed list of your responses to the review comments and a description of the changes you have made in the manuscript.

(2) Two versions of the manuscript: one with either highlights or tracked changes denoting where the text has been changed (uploaded as a "Revised Article with Changes Highlighted" file); the other a clean version (uploaded as the article file).

(3) If available, a striking still image (a new image if one is available or an existing one from within your manuscript). If your manuscript is accepted for publication, this image may be featured on our website. Images should ideally be high resolution, eye-catching, single panel images; where one is available, please use 'add file' at the time of resubmission and select 'striking image' as the file type. 

Please provide a short caption, including credits, uploaded as a separate "Other" file. If your image is from someone other than yourself, please ensure that the artist has read and agreed to the terms and conditions of the Creative Commons Attribution License at http://journals.plos.org/plosntds/s/content-license (NOTE: we cannot publish copyrighted images). 

(4) If applicable, we encourage you to add a list of accession numbers/ID numbers for genes and proteins mentioned in the text (these should be listed as a paragraph at the end of the manuscript). You can supply accession numbers for any database, so long as the database is publicly accessible and stable. Examples include LocusLink and SwissProt.

(5) To enhance the reproducibility of your results, we recommend that you deposit your laboratory protocols in protocols.io, where a protocol can be assigned its own identifier (DOI) such that it can be cited independently in the future. For instructions see http://journals.plos.org/plosntds/s/submission-guidelines#loc-methods

While revising your submission, please upload your figure files to the Preflight Analysis and Conversion Engine (PACE) digital diagnostic tool, https://pacev2.apexcovantage.com/ PACE helps ensure that figures meet PLOS requirements. To use PACE, you must first register as a user. Then, login and navigate to the UPLOAD tab, where you will find detailed instructions on how to use the tool. If you encounter any issues or have any questions when using PACE, please email us at figures@plos.org.

We hope to receive your revised manuscript by Oct 04 2019 11:59PM. If you anticipate any delay in its return, we ask that you let us know the expected resubmission date by replying to this email.

To submit a revision, go to https://www.editorialmanager.com/pntd/ and log in as an Author. You will see a menu item call Submission Needing Revision. You will find your submission record there. 

Sincerely,

Syamal Roy

Guest Editor

Charles Jaffe

Deputy Editor

Reviewer's Responses to Questions

**Key Review Criteria Required for Acceptance?**

**Methods**

-Are the objectives of the study clearly articulated with a clear testable hypothesis stated?

-Is the study design appropriate to address the stated objectives?

-Is the population clearly described and appropriate for the hypothesis being tested?

-Is the sample size sufficient to ensure adequate power to address the hypothesis being tested?

-Were correct statistical analysis used to support conclusions?

-Are there concerns about ethical or regulatory requirements being met?

Reviewer #1: The studies by Abu Musa et al are of interest. Their findings help reconcile discrepant observations in the literature.

Reviewer #2: The objectives of the study are clearly defined.

The study design is appropriate, but there are problems in the interpretation of data.

The same previous answer 

The number of experiments and mice used in the experiments is fine.

The authors used ANOVA, but no test for normality of the data was done

No concerns about ethical or regulatory requirements

Reviewer #3: Although the methods are suitable, can the authors please clarify/rectify the following points? 

 1) In regards to Fig 1A, Fig 2A and S3A Fig, why was one-way ANOVA used to determine statistical significance? Since their phenotype depends time and experimental condition, two-way ANOVA seems more appropriate. 

 2) In Figs 1A and 2A, errors bars are too thick, which makes it confusing for readers to properly visualize and assess the data. Since there are four different time series in the same graphs, the authors should use different colours to better differentiate the lines.

 3) Numerical data in Figs 3-5 and S5 Fig should also be depicted in the form of scatter plots, as done in Fig 1C, S1 Fig and elsewhere.

**Results**

-Does the analysis presented match the analysis plan?

-Are the results clearly and completely presented?

-Are the figures (Tables, Images) of sufficient quality for clarity?

Reviewer #1: Overall data and conclusions are clearly presented and discussed.

Reviewer #2: Yes, it does.

The Figures presented low quality, it is not clear the statistical differences shown in the Figures

Reviewer #3: 1) Overall, results are laid out well, but the description of the data is often incorrect (see ‘Summary and General Comments’). For example, Figs 1-2 do not really show a ‘T lymphocyte-dependent pathogenesis by L. major Friedlin and independent in 5ASKH’, but rather whether lymphocytes worsen/help control disease by those strains. 

 2) In Fig 1B, the photos pertaining to the 5ASKH strain do not recapitulate the data shown in 1A, which show that 5ASKH-induced lesions worsened in BALB/c nude mice. Since the photos in 1B show the opposite, perhaps the authors accidentally flipped them?

**Conclusions**

-Are the conclusions supported by the data presented?

-Are the limitations of analysis clearly described?

-Do the authors discuss how these data can be helpful to advance our understanding of the topic under study?

-Is public health relevance addressed?

Reviewer #1: The paragraph beginning on line 361 discusses observations with the different Leishmania species; however, the authors may consider including a bit more information about the pedigree of both of the strains that were tested, in other to give some context of the diversity of disease presentations that may be seen in L. major endemic regions.

Reviewer #2: The conclusions are supported by the data, but the discussion is a summary of the results.

The limitations of analysis are not described.

Yes, they do, but some of the statements are not in agreement with the literature,

Yes, it is

Reviewer #3: The conclusions are partially supported by the authors’ findings. Section 5 (Conclusions) should focus on the positive and negative influence that lymphocytes have on the pathologies caused by the strains that the authors employed. As such, that section should be written concisely. The text pertaining to parasite virulence factors and whatnot should be integrated within the main body of the Discussion (section 4).

**Editorial and Data Presentation Modifications?**

Reviewer #1: No concerns noted

Reviewer #2: (No Response)

Reviewer #3: 1) Lines 379-392 ramble too much about the link that their mouse-based work has with Leishmania/HIV co-infected patients. In fact, lines 387-392 contain a very speculative and incoherent argument.

 2) Authors should employ current and proper genetic terminology to describe their mouse strains throughout the manuscript. 

 3) In the title, the word ‘the’ should be omitted. As per my major concerns (outlined in the ‘Summary and General Comments’ section), the word ‘requirement’ should be replaced by ‘influence’. 

I suggest the following title: ‘Lymphocytes influence Leishmania major pathogenesis in a strain-dependent manner’.

**Summary and General Comments**

Reviewer #1: The results of the study are significant as they bring to the fore observations that cutaneous disease presentation in endemic areas is variable. The variability could apparently be be due to strain differences within parasite species that impact immune responses.

Reviewer #2: The manuscript showed differences in the development of lesions by two different strains of L. major (Friedlin and 5ASKH) using different mouse strains such as BALB/c and nude BALB/c, as well as C57Bl/6, gene 2 (Rag2) knockout (KO) C57BL/6, and NOD-SCID mice. The authors observed that in mice lacking T/B responses, 5ASKH developed a larger lesion, with ulceration whereas Friedlin strain induced in these mice smaller lesions, with less inflammation.They also performed experiments employing flow cytometry, and there was a large recruitment of cells to the lesion caused by 5ASKH compared to Friedlin. Concerning cytokines, the results were similar, with an increase in IFN-g, IL-10, TNF and IL-12p40 in the cells from lesions induced by 5ASKH. One intriguing fact is that there was no statistical difference in the parasite load from the lesions caused by the two different strains. Although, the results are interesting, this observation of development of pathology caused by Leishmania has been discussed for a long time, as the author stated, and the results of this manuscript reinforce this idea.

However, there are some aspects that deserve discussion:

1-There was no explanation about the assay to calculate the parasite load.

2- the amount of cells collected from the footpad of infected mouse. Is this material also used to evaluate the parasite load?

3- Which type of response was observed in the draining lymph nodes of the mice infected by the two strains of L. major?

4- The authors did not analyze the inflammatory responses before 6 weeks or 4 weeks of infection in some experiments. It would be interesting to perform these experiments, comparing with the wild type mouse.

5-One important aspect to discuss, is related to the control of L. major Friedlin in the lesion, since the recruitment of cells is really low and also, the inflammatory cytokines able to activate the macrophages to eliminate the parasite.

6- Important data that are not present in the manuscript is related with the source of cytokines in the mice without T/B. Which cell is producing IFN-g?

7- The Discussion needs improvement. It is a summary of results without any discussion. The conclusions are really repetitive.

Reviewer #3: The study by Abu Musa et al. sought to elucidate how parasite strains influence the role of lymphocytes during Leishmania major infection. To this end, the authors used two strains of differing virulence (Friedlin and 5ASKH) to infect mice that contained or lacked lymphocytes.

Although interesting, I have significant concerns regarding how the authors interpreted their data. 

 1) Throughout the manuscript, the authors repeatedly claim that while the pathogenesis caused by the Friedlin strain is dependent on lymphocytes, that of the 5ASKH strain is largely not. The authors gone on the argue that the **requirement** for lymphocytes in pathogenesis is strain-dependent. Nonetheless, these statements are not really backed by the data. Instead of portraying a requirement for lymphocytes, the data in Figs 1A and 2A clearly show that lymphocytes influence (either exacerbate or diminish) infection outcome depending on parasite strain. This is further nuanced by the time-dependent development of the lesions, which should be more explicitly described in the results section. 

 2) The finding that lymphocyte presence helps control 5ASKH infection does not mean that pathogenicity is independent of lymphocytes (or the opposite in the case of the Friedlin strain). In order to prove that there really is a requirement for those cells, authors must perform adoptive cell transfer experiments (eg. introduce WT lymphocytes into nude/RAG2 mice or vice-versa).

PLOS authors have the option to publish the peer review history of their article (what does this mean?). If published, this will include your full peer review and any attached files.

Reviewer #1: No

Reviewer #2: No

Reviewer #3: No

---

## [Decision Letter · Decision Letter 1]

3 Oct 2019

Dear Prof. Hamano:

Thank you very much for submitting your manuscript "Lymphocytes influence Leishmania major pathogenesis in a strain-dependent manner" (PNTD-D-19-01017R1) for review by PLOS Neglected Tropical Diseases. Your manuscript was fully evaluated at the editorial level and by independent peer reviewers. The reviewers appreciated the attention to an important topic but identified some aspects of the manuscript that should be improved.

We therefore ask you to modify the manuscript according to the review recommendations before we can consider your manuscript for acceptance. Your revisions should address the specific points made by each reviewer.

(1) A letter containing a detailed list of your responses to the review comments and a description of the changes you have made in the manuscript.

(2) Two versions of the manuscript: one with either highlights or tracked changes denoting where the text has been changed (uploaded as a "Revised Article with Changes Highlighted" file ); the other a clean version (uploaded as the article file).

(3) If available, a striking still image (a new image if one is available or an existing one from within your manuscript). If your manuscript is accepted for publication, this image may be featured on our website. Images should ideally be high resolution, eye-catching, single panel images; where one is available, please use 'add file' at the time of resubmission and select 'striking image' as the file type. 

Please provide a short caption, including credits, uploaded as a separate "Other" file. If your image is from someone other than yourself, please ensure that the artist has read and agreed to the terms and conditions of the Creative Commons Attribution License at http://journals.plos.org/plosntds/s/content-license (NOTE: we cannot publish copyrighted images). 

(4) Appropriate Figure Files 

Please remove all name and figure # text from your figure files upon submitting your revision. Please also take this time to check that your figures are of high resolution, which will improve both the editorial review process and help expedite your manuscript's publication should it be accepted. Please note that figures must have been originally created at 300dpi or higher. Do not manually increase the resolution of your files. For instructions on how to properly obtain high quality images, please review our Figure Guidelines, with examples at: http://journals.plos.org/plosntds/s/figures

While revising your submission, please upload your figure files to the Preflight Analysis and Conversion Engine (PACE) digital diagnostic tool, https://pacev2.apexcovantage.com/ PACE helps ensure that figures meet PLOS requirements. To use PACE, you must first register as a user. Then, login and navigate to the UPLOAD tab, where you will find detailed instructions on how to use the tool. If you encounter any issues or have any questions when using PACE, please email us at figures@plos.org.

We hope to receive your revised manuscript by Dec 02 2019 11:59PM. If you anticipate any delay in its return, we ask that you let us know the expected resubmission date by replying to this email.

To submit your revised files, please log in to https://www.editorialmanager.com/pntd/

Sincerely,

Syamal Roy

Guest Editor

Charles Jaffe

Deputy Editor

Please respond to the minor comments by reviewer 3. Thank you!

Reviewer's Responses to Questions

**Key Review Criteria Required for Acceptance?**

**Methods**

-Are the objectives of the study clearly articulated with a clear testable hypothesis stated?

-Is the study design appropriate to address the stated objectives?

-Is the population clearly described and appropriate for the hypothesis being tested?

-Is the sample size sufficient to ensure adequate power to address the hypothesis being tested?

-Were correct statistical analysis used to support conclusions?

-Are there concerns about ethical or regulatory requirements being met?

Reviewer #2: Yes

Reviewer #3: The authors have properly addressed my comments in regards to methodology.

**Results**

-Does the analysis presented match the analysis plan?

-Are the results clearly and completely presented?

-Are the figures (Tables, Images) of sufficient quality for clarity?

Reviewer #2: Yes

Reviewer #3: Overall, the layout and description of results have been improved.

**Conclusions**

-Are the conclusions supported by the data presented?

-Are the limitations of analysis clearly described?

-Do the authors discuss how these data can be helpful to advance our understanding of the topic under study?

-Is public health relevance addressed?

Reviewer #2: Yes

Reviewer #3: Although the discussion and conclusion sections have been enhanced, the authors can still trim off a lot of experimental detail that is already presented in the results and figure legends.

**Editorial and Data Presentation Modifications?**

Reviewer #2: The authors answered all the questions raised up by the reviewer including new experiments that improve the study and the manuscript.

Reviewer #3: Prior to formal acceptance, the authors should thoroughly proofread their manuscript to correct errors such as the ones found on line 45 (may induced -> may induce), line 349 (were not analyzed -> was not analyzed), line 385 (may caused -> may have caused), etc. […]

The graphs appear to be blurry in the PDF version. Authors should ensure that their GraphPad-generated images have been exported in TIFF format at 600dpi, and that their multi-panel montages are saved in a non lossy compression format.

**Summary and General Comments**

Reviewer #2: The authors answered all the questions raised up by the reviewer including new experiments that improve the study and the manuscript.

Reviewer #3: In this revised submission, Abu Musa and colleagues have addressed most of my concerns. Prospective immunological studies are needed to decipher the differential lymphocyte response towards different Leishmania strains and species.

PLOS authors have the option to publish the peer review history of their article (what does this mean?). If published, this will include your full peer review and any attached files.

Reviewer #2: No

Reviewer #3: Yes: Guillermo Arango Duque

---

## [Editor Report · Decision Letter 2]

22 Oct 2019

Dear Prof. Hamano,

We are pleased to inform you that your manuscript, "Lymphocytes influence Leishmania major pathogenesis in a strain-dependent manner", has been editorially accepted for publication at PLOS Neglected Tropical Diseases.

Before your manuscript can be formally accepted and sent to production you will need to complete our formatting changes, which you will receive in a follow up email. Please note: your manuscript will not be scheduled for publication until you have made the required changes.

IMPORTANT NOTES

* Copyediting and Author Proofs: To ensure prompt publication, your manuscript will NOT be subject to detailed copyediting and you will NOT receive a typeset proof for review. The corresponding author will have one final opportunity to correct any errors when sent the requests mentioned above. Please review this version of your manuscript for any errors.

* If you or your institution will be preparing press materials for this manuscript, please inform our press team in advance at plosntds@plos.org. If you need to know your paper's publication date for media purposes, you must coordinate with our press team, and your manuscript will remain under a strict press embargo until the publication date and time. PLOS NTDs may choose to issue a press release for your article. If there is anything that the journal should know, please get in touch.

*Now that your manuscript has been provisionally accepted, please log into EM and update your profile. Go to http://www.editorialmanager.com/pntd, log in, and click on the "Update My Information" link at the top of the page. Please update your user information to ensure an efficient production and billing process.

*Note to LaTeX users only - Our staff will ask you to upload a TEX file in addition to the PDF before the paper can be sent to typesetting, so please carefully review our Latex Guidelines [http://www.plosntds.org/static/latexGuidelines.action] in the meantime.

Best regards,

Syamal Roy

Guest Editor

Charles Jaffe

Deputy Editor

---

## [Editor Report · Acceptance letter]

8 Nov 2019

Dear Prof. Hamano,

We are delighted to inform you that your manuscript, "Lymphocytes influence *Leishmania major* pathogenesis in a strain-dependent manner," has been formally accepted for publication in PLOS Neglected Tropical Diseases.

Best regards,

Serap Aksoy

Editor-in-Chief

Shaden Kamhawi

Editor-in-Chief
